# Quantum Uncertainties and Holism Seem to Render Irrelevant Qudit-Semantics

**DOI:** 10.3390/e23060735

**Published:** 2021-06-10

**Authors:** Roberto Leporini

**Affiliations:** Department of Economics, University of Bergamo, Via dei Caniana, 2, I-24127 Bergamo, Italy; roberto.leporini@unibg.it

**Keywords:** quantum logics, quantum gates, holistic semantics

## Abstract

We consider a semantics based on the peculiar holistic features of the quantum formalism. Any formula of the language gives rise to a quantum circuit that transforms the density operator associated to the formula into the density operator associated to the atomic subformulas in a reversible way. The procedure goes from the whole to the parts against the compositionality-principle and gives rise to a semantic characterization for a new form of quantum logic that has been called “Łukasiewicz quantum computational logic”. It is interesting to compare the logic based on qubit-semantics with that on qudit-semantics. Having in mind the relationships between classical logic and Łukasiewicz-many valued logics, one could expect that the former is stronger than the fragment of the latter. However, this is not the case. From an intuitive point of view, this can be explained by recalling that the former is a very weak form of logic. Many important logical arguments, which are valid either in Birkhoff and von Neumann’s quantum logic or in classical logic, are generally violated.

## 1. Introduction

Quantum computational logics are new forms of quantum logics inspired by the theory of quantum computation [1]. While sharp and unsharp quantum logics refer to possible structures of physical events [2], the basic objects of quantum computational logics are pieces of quantum information. The simplest piece of quantum information is a *qubit* (state): a unit-vector of the Hilbert space $C^{2}$ that can be represented as a superposition $|\psi \rangle =c_{0}|0\rangle +c_{1}|1\rangle$. The two basis elements (|0〉 and |1〉) represent the two classical truth-values or, equivalently, the classical bits. The simplest generalization of qubits are represented by *qutrits*: unit-vectors living in a space $C^{3}$. In the many-valued generalization, one can consider *d* truth-values and many-valued connectives which allow one to get logical truths differently from the qubit-case [3].

The most natural semantics is a form of *holistic semantics*, based on the peculiar holistic features of the quantum theory. Any formula of the language gives rise to a quantum circuit that transforms in a reversible way the density operator associated to the formula into the density operator associated to the atomic subformulas [1]. The procedure goes from the whole to the parts and not the other way around, against the compositionality-principle. This gives rise to a natural semantic characterization for a new form of quantum logic that has been called “Łukasiewicz quantum computational logic ” (**ŁQCL**). In this semantics, the entropy of a formula is the average level of information or uncertainty inherent in the possible truth-values. Indeed, the truth-value associated to a formula is the expected value. As expected, the language of **ŁQCL** is richer than the language of “Holistic quantum computational logic” (**HQCL**), since some basic connectives (like negation and conjunction) turn out to be split into different kinds of connectives. It is interesting to compare the logic **HQCL** with the fragment **ŁQCL*** of **ŁQCL** whose formulas are expressed in the language of **HQCL**. In [4], we showed, as expected, that the compositional logic characterized by the qubit semantics is stronger than the compositional Łukasiewicz quantum computational logic by a counterexample. The holistic case is a different matter because quantum uncertainties and holism play a fundamental role.

The paper is organized as follows. In Section 2, we recall some basic notions of quantum computational logics and the qudit-semantics based on the Bertini gate and on genuine quantum gates. In Section 3, we show some new results that are useful for proving that **HQCL** and the fragment **ŁQCL*** characterize the same logic. Finally, the conclusion is drawn in Section 4.

## 2. Basic Notions

Let us first recall some basics useful for quantum computational logics [5,6,7,8,9]. As is well known, the general mathematical environment is the Hilbert space $H_{d}^{\left(n\right)}:=\underset{n-times}{\underset{︸}{C^{d}⊗\ldots ⊗C^{d}}}$ (*n*-fold tensor product where d≥2 and n≥1) with the canonical orthonormal basis $B^{\left(n\right)}$:$$B^{\left(n\right)}=\left\{|x_{1},\ldots ,x_{n}\rangle :x_{1}\in D,\ldots ,x_{n}\in D\right\},$$
where $D=\left\{0,\frac{1}{d-1},\ldots ,1\right\}$, |0〉=100∫x⋮0,|1d−1〉=010∫x⋮0,…,|1〉=00∫x⋮01, while $|x_{1},\ldots ,x_{n}\rangle$ is an abbreviation for $|x_{1}\rangle ⊗\ldots ⊗|x_{n}\rangle$.

For instance, the truth-values of the qutrit space $C^{3}$ will be:$$\left|0\rangle =\right|\frac{0}{2}\rangle =\left(\begin{array}{c} 1\\ 0\\ 0 \end{array}\right),|\frac{1}{2}\rangle =\left(\begin{array}{c} 0\\ 1\\ 0 \end{array}\right),|1\rangle =|\frac{2}{2}\rangle =\left(\begin{array}{c} 0\\ 0\\ 1 \end{array}\right).$$

Any piece of quantum information is represented by a density operator ρ of $H_{d}^{\left(n\right)}$. A *quregister* (state) is represented by a unit-vector |ψ〉 which is a pure state of $H_{d}^{\left(n\right)}$ or, equivalently, by the corresponding projection-operator $P_{|\psi \rangle }$ that projects over the closed subspace determined by |ψ〉. Following a standard convention, $P_{|0\rangle }$ represents the *Falsity*, $P_{|1\rangle }$ represents the *Truth* and $P_{|\frac{j}{d-1}\rangle }$ represents an *intermediate* truth-value, where 0<j<d−1. In this framework, a *truth-value projection* $P_{\frac{j}{d-1}}^{\left(n\right)}$ of $H_{d}^{\left(n\right)}$ is a projection whose range is the closed subspace spanned by all quregisters ending with $\frac{j}{d-1}$, where 0≤j≤d−1.

By applying the Born rule, one can compute the probability that the information stocked by the density operator ρ is the truth-value $\frac{j}{d-1}$:$$p_{\frac{j}{d-1}}\left(\rho \right)=\mathrm{tr}\left(\rho \;P_{\frac{j}{d-1}}^{\left(n\right)}\right),$$
where tr is the trace-functional and 0≤j≤d−1. On this basis, one can define the probability for any density operator ρ of $H_{d}^{\left(n\right)}$ as the expected truth-value.

**Definition** **1.**
*The probability of a density operator.*
$$p\left(\rho \right)=\frac{1}{d-1}\sum_{j=1}^{d-1}j\;p_{\frac{j}{d-1}}\left(\rho \right)$$


Trivially, we have:$$p\left(\rho \right)=\mathrm{tr}\left(\rho \;(I^{\left(n-1\right)}⊗E)\right),$$
where $I^{\left(n-1\right)}$ is the identity operator of $H^{\left(n-1\right)}$ and *E* is the effect of the form
$$\left[\begin{array}{ccccc} 0 & 0 & 0 & \cdots & 0\\ 0 & \frac{1}{d-1} & 0 & \cdots & 0\\ 0 & 0 & \frac{2}{d-1} & \cdots & \vdots \\ \vdots & \vdots & \vdots & \ddots & 0\\ 0 & 0 & \cdots & 0 & 1 \end{array}\right]$$
For instance, $p\left(P_{c_{0}|0\rangle +c_{1}|1\rangle }\right)={\left|c_{1}\right|}^{2}$.

In the qudit-framework, basis elements represent classical pieces of information, although based on many-valued systems of truth-values. At the same time, the probabilistic behavior is generally different: in the qubit-case, probabilities of basis elements are dichotomic, while, in the qudit-case, there are registers $|x_{1},\ldots ,x_{n}\rangle$ such that $p\left(P_{|x_{1},\ldots ,x_{n}\rangle }\right)\neq 0,1$. A typical example is represented by $|\frac{1}{2}\rangle$, where $p\left(P_{|\frac{1}{2}\rangle }\right)=\frac{1}{2}.$ Thus, in qudit-spaces “classical” pieces of information may have an indeterminate probability value. Of course, when the truth-value number *d* is greater than 2, one takes into account the characteristic “many-valued features” of the space $C^{d}$.

Consider the product-space
$$H^{\left(m+n\right)}=H^{\left(m\right)}⊗H^{\left(n\right)}.$$
Any density operator ρ of $H^{\left(m+n\right)}$ represents a bipartite state for a composite physical system $S=S_{1}+S_{2}$. According to the quantum theory, ρ determines the *reduced states*
$Red_{\left[m,n\right]}^{\left(1\right)}\left(\rho \right)$ and $Red_{\left[m,n\right]}^{\left(2\right)}\left(\rho \right)$ that represent the states of $S_{1}$ and $S_{2}$, respectively. The notion of reduced state can be naturally defined for multi-partite systems. Let $S=S_{1}+\ldots +S_{n}$ be a composite system whose Hilbert space is the product-space. Any state ρ of *S* determines the reduced state $Red^{\left(j\right)}\left(\rho \right)$ of each subsystem $S_{j}$.

Matrix bases can be used to decompose density matrices associated to states of quantum systems. In the qubit-case, an important basis is formed by the identity matrix and by the three Pauli matrices. A density matrix can be expressed by a 3-dimensional vector, the Bloch vector, that lies within the Poincaré–Bloch ball (sphere of radius 1) [10]. In higher dimensions, two bases play an important role: the generalized Pauli basis and the Weyl operator basis. For any j,k,l such that $1\leq j\leq d^{2}-1$, 0≤k<l≤d−1, the generalized Pauli matrices $\sigma_{j}$ on $C^{d}$ can be defined as follows: $$\sigma_{j}=\left\{\begin{array}{c} |\frac{k}{d-1}\rangle \langle \frac{l}{d-1}|+|\frac{l}{d-1}\rangle \langle \frac{k}{d-1}|\\ \;\mathrm{if}j\leq \frac{d(d-1)}{2}\mathrm{and}j=\frac{k(1-k)}{2}+\left(d-2\right)k+l;\\ -i|\frac{k}{d-1}\rangle \langle \frac{l}{d-1}|+i|\frac{l}{d-1}\rangle \langle \frac{k}{d-1}|\\ \;\mathrm{if}\frac{d(d-1)}{2}<j\leq d\left(d-1\right)\mathrm{and}j=\frac{d(d-1)+k(1-k)}{2}+\left(d-2\right)k+l;\\ \sqrt{\frac{2}{l(l+1)}}\left(\sum_{k=0}^{l-1}|\frac{k}{d-1}\rangle \langle \frac{k}{d-1}|-l|\frac{l}{d-1}\rangle \langle \frac{l}{d-1}|\right)\\ \;\mathrm{if}j>d(d-1)\mathrm{and}j=d(d-1)+l. \end{array}\right.$$
They are the standard SU(d) generators. The expansion of ρ with respect to the orthogonal basis $\left\{I^{\left(1\right)},\sigma_{j}:1\leq j\leq d^{2}-1\right\}$ is
$$\rho =\frac{1}{d}\left(I^{\left(1\right)}+\sqrt{\frac{d(d-1)}{2}}\sum_{j=1}^{d^{2}-1}b_{j}\sigma_{j}\right),$$
where $b_{j}=\sqrt{\frac{d}{2(d-1)}}\mathrm{tr}\left(\rho \;\sigma_{j}\right)\in R$.

$b=(b_{1},\ldots ,b_{d^{2}-1})$ represents the Bloch vector associated to ρ with respect to the basis $\left\{I^{\left(1\right)},\sigma_{j}:1\leq j\leq d^{2}-1\right\}$, that lies within the hypersphere of radius 1. The Bloch vector has real components that can be obtained as expectation values of measurable quantities. When d=3, we obtain the Gell–Mann Hermitian matrices and the Bloch vector can be obtained as expectation values of spin 1 operators.

Let us recall what happens in the semantics of Łukasiewicz’ logics which represent special examples of fuzzy logics. In this semantics, the set of truth-values is identified either with the real interval [0,1] or with a finite subset thereof (D), the diametrical negation is defined like in the classical case (¬x:=1−x). At the same time, the conjunction is split into two different irreversible operations, the Łukasiewicz-conjunction (x⊙y:=max(x+y−1,0)) and the lattice-conjunction (x⊓y:=min(x,y), also called min-conjunction). As expected, two different kinds of inclusive disjunctions can be defined via the de Morgan-law: x⊕y:=¬(¬x⊙¬y)=min(x+y,1);x⊔y:=¬(¬x⊓¬y)=max(x,y). While ⊙ and ⊓ are the same operation in the two-valued semantics, when d>2 these two conjunctions satisfy different semantic properties. The Łukasiewicz-conjunction is generally non-idempotent (x⊙x≠x). The lattice-conjunction gives rise to possible violations of the non-contradiction principle (x⊓¬x≠0), as so happens in the case of most fuzzy logics whose basic aim is modeling ambiguous and unsharp semantic situations. At the same time, ⊓ behaves as a lattice-operation in the truth-value partial order.

Following Zawirski [11] or Chang [12,13], the Łukasiewicz approach to many-valued logics can be recovered on the basis of {⊕,¬}, or equivalently {⊙,¬}. We also consider two modal connectives: possibility (⋄x:= 0 if x=0; 1 otherwise) and necessity (□x:=¬⋄¬x). In addition to the diametrical negation, other negation connectives can be defined: the intuitionistic negation (∼x:=¬⋄x) and the anti-intuitionistic negation (also contingency, ♭x:=⋄¬x). All these logical operations can be simulated by convenient reversible gates. Pure pieces of quantum information are transformed by *quantum logical gates* (briefly, *gates*) that play a special role from the logical point of view [14]. In this paper, we will focus on the *Bertini gate* that represents a semiclassical gate, because it always transforms basis elements representing classical information into basis elements.

**Definition** **2.**
*The Bertini gate.*

*The Bertini gate is the linear operator B such that for every element |x,y,z〉 of the computational basis $B^{\left(3\right)}$:*
$$B(|x,y,z\rangle )=\left\{\begin{array}{cc} |1-x,x,x+y-1\rangle & \mathrm{if}\;z=0\;\mathrm{and}\;x+y>1\\ |y,x+z,0\rangle & \mathrm{if}\;x=1-y\;\mathrm{and}\;0<z\leq y\\ |y-z,z,z\rangle & \mathrm{if}\;x=0\;\mathrm{and}\;z<y<1\\ |0,x+y,z\rangle & \mathrm{if}\;y=z\;\mathrm{and}\;0<x<1-y\\ |1,0,0\rangle & \mathrm{if}\;x=0,y=1\;\mathrm{and}\;z=0\\ |0,1,0\rangle & \mathrm{if}\;x=1,y=0\;\mathrm{and}\;z=0\\ |x,y,z\rangle & \mathrm{otherwise} \end{array}\right.$$


The Bertini gate is both conservative (i.e., the sum of input values is preserved into the output) and self-reversible (i.e., $B\;B=I^{\left(3\right)}$), and it has the following truth table for three values (see Table 1).

The Bertini gate is functional complete: by fixing the values of a subset of their inputs, one can get a set of connectives that, equipped with a suitable set of intermediate logical constants, are able to realize any mapping from ${\left\{0,\frac{1}{d-1},\ldots ,1\right\}}^{n}$ to $\left\{0,\frac{1}{d-1},\ldots ,1\right\}$. As is well known, Łukasiewicz logics without such constants is incomplete: for instance, it cannot express the function which is identically equal to $\frac{1}{d-1}$. In [15] Aharonov showed a simple proof that Hadamard and Toffoli are an approximately universal set of quantum gates. A similar result can be obtained by considering the Hadamard gate and the Bertini gate.

An interesting application of quantum logics to the formal verification of protocols in quantum computation and communication was developed by Smets et al. [16]. In particular, they make use of the probabilistic logic of quantum programs to provide a formal specification of the quantum voting protocol for anonymous surveying with its correctness. Applications to quantum key distribution protocol and to quantum leader election protocol, that aims to randomly select a leader in a group of agents, are shown in [17].

Consider the product-space $H^{\left(3\right)}=H^{\left(1\right)}⊗H^{\left(1\right)}⊗H^{\left(1\right)}.$ Any density operator ρ of $H^{\left(3\right)}$ represents $S=S_{1}+S_{2}+S_{3}$. According to the quantum formalism, a possible state ρ for the composite physical system determines the *reduced states*
$Red^{\left(1\right)}\left(\rho \right)$, $Red^{\left(2\right)}\left(\rho \right)$ and $Red^{\left(3\right)}\left(\rho \right)$, respectively. In such a case, ρ represents a *tripartite state* with respect to the decomposition [1,1,1]. In particular,
$$Red^{\left(2\right)}\left(P_{B(|0,y,z\rangle )}\right)=P_{|y⊓z\rangle }$$
$$Red^{\left(3\right)}\left(P_{B(|x,y,0\rangle )}\right)=P_{|x⊙y\rangle }$$
$$Red^{\left(1\right)}\left(P_{B(|x,1,0\rangle )}\right)=P_{|\neg x\rangle }$$
$$Red^{\left(3\right)}\left(P_{B(|x,1,1\rangle )}\right)=P_{|⋄x\rangle }$$

Quantum computational logics are involved in the language last qudit-target-gates [18]. One may think this is a limitation due to the fact that the target may be in another position, but it is always possible to exchange the target to the last position using a swap gate which plays a central role in network designs for quantum computation.

**Definition** **3.**
*The swap gate.*

*For any n≥2, j≥1, k≤n such that j<k, the swap gate is the linear operator ${\mathrm{Swap}}_{j,k}^{\left(n\right)}$ such that for every element $|x_{1},\ldots ,x_{n}\rangle$ of the computational basis $B^{\left(n\right)}$:*
${\mathrm{Swap}}_{j,k}^{\left(n\right)}(|x_{1},\ldots ,x_{j},\ldots ,x_{k},\ldots ,x_{n}\rangle )=|x_{1},\ldots ,x_{k},\ldots ,x_{j},\ldots ,x_{n}\rangle$


Therefore, with a single gate, a Łukasiewicz conjunction ${\mathrm{And}}_{Ł}$ and a lattice-conjunction And can be defined for any density operator ρ in $H^{\left(m+n\right)}$:$${\mathrm{And}}_{Ł}\left(\rho \right):=\left\{\begin{array}{cc} & B\;(\rho ⊗P_{|0\rangle })B\;\mathrm{if}\;m=n=1;\\ & \left(I^{\left(m+n-2\right)}⊗\;B\right)\;\left(\rho ⊗P_{|0\rangle }\right)\left(I^{\left(m+n-2\right)}⊗\;B\right)\;\mathrm{otherwise}. \end{array}\right.$$
$$\mathrm{And}\left(\rho \right):=\left\{\begin{array}{cc} & {\mathrm{Swap}}_{2,3}^{\left(3\right)}\;B\;{\mathrm{Swap}}_{1,3}^{\left(3\right)}\;\left(\rho ⊗P_{|0\rangle }\right){\mathrm{Swap}}_{1,3}^{\left(3\right)}\;B\;{\mathrm{Swap}}_{2,3}^{\left(3\right)}\;\mathrm{if}\;m=n=1;\\ & {\mathrm{Swap}}_{m+n,m+n+1}^{\left(m+n+1\right)}\;\left(I^{\left(m+n-2\right)}⊗B\right)\;{\mathrm{Swap}}_{m+n-1,m+n+1}^{\left(m+n+1\right)}\;{\mathrm{Swap}}_{m,m+n-1}^{\left(m+n+1\right)}\;\left(\rho ⊗P_{|0\rangle }\right)\\ & {\mathrm{Swap}}_{m,m+n-1}^{\left(m+n+1\right)}\;{\mathrm{Swap}}_{m+n-1,m+n+1}^{\left(m+n+1\right)}\;\left(I^{\left(m+n-2\right)}⊗B\right)\;{\mathrm{Swap}}_{m+n,m+n+1}^{\left(m+n+1\right)}\;\mathrm{otherwise}. \end{array}\right.$$

Moreover, a possibility and a negation can be defined for any density operator ρ in $H^{\left(n\right)}$:$$\mathrm{Pos}\left(\rho \right):=\left\{\begin{array}{cc} & B\;(\rho ⊗P_{|11\rangle })\;B\;\mathrm{if}\;n=1;\\ & \left(I^{\left(n-1\right)}⊗B\right)\;\left(\rho ⊗P_{|11\rangle }\right)\left(I^{\left(n-1\right)}⊗B\right)\;\mathrm{otherwise}. \end{array}\right.$$
$$\mathrm{Not}\left(\rho \right):=\left\{\begin{array}{cc} & {\mathrm{Swap}}_{1,3}^{\left(3\right)}\;B\;\left(\rho ⊗P_{|10\rangle }\right)\;B\;{\mathrm{Swap}}_{1,3}^{\left(3\right)}\;\mathrm{if}\;n=1;\\ & {\mathrm{Swap}}_{n,n+2}^{\left(n+2\right)}\;\left(I^{\left(n-1\right)}⊗B\right)\;\left(\rho ⊗P_{|10\rangle }\right)\left(I^{\left(n-1\right)}⊗B\right){\mathrm{Swap}}_{n,n+2}^{\left(n+2\right)}\;\mathrm{otherwise}. \end{array}\right.$$

In [14], we showed some interesting relations between the probability function p and the connectives that are useful in the synthesis and simplification of many-valued logic digital circuits. In fact, the probability of the gates can be described in terms of the corresponding logical operation (⊙,⊓) and continuous t-norms (⊕,·).

The following gate represents a generalization of the Hadamard gate of $C^{2}$.

**Definition** **4.**
*The Hadamard gate on $H_{d}^{\left(1\right)}$*

*The Hadamard gate on the space $H_{d}^{\left(1\right)}$ is the linear operator ${\sqrt{I}}^{\left(1\right)}$ that satisfies the following condition for every element |x〉 of the canonical basis:*
$${\sqrt{I}}^{\left(1\right)}|x\rangle =\frac{1}{\sqrt{2}}(c|x\rangle +|1-x\rangle ),$$
*where $c=\left\{\begin{array}{cc} 1, & \mathrm{if}\;x<\frac{1}{2};\\ \sqrt{2}-1, & \mathrm{if}\;x=\frac{1}{2};\\ -1, & \mathrm{if}\;x>\frac{1}{2}. \end{array}\right.$*


As happens in the qubit-case, ${\sqrt{I}}^{\left(1\right)}$ transforms each basis element into an equal superposition of the basis element and of its negation. Moreover, it is a square root of the identity: ${\sqrt{I}}^{\left(1\right)}{\sqrt{I}}^{\left(1\right)}=I^{\left(1\right)}.$

For instance, consider the qutrit-space $C^{3}$. We have:${\sqrt{I}}^{\left(1\right)}|\frac{0}{2}\rangle =\frac{1}{\sqrt{2}}\left(|\frac{0}{2}\rangle \;\;+|1-\frac{0}{2}\rangle \right)=\frac{1}{\sqrt{2}}\left(|\frac{0}{2}\rangle \;\;+|\frac{2}{2}\rangle \right)=\frac{1}{\sqrt{2}}\left(|0\rangle \;\;+|1\rangle \right)$;${\sqrt{I}}^{\left(1\right)}|\frac{1}{2}\rangle =\frac{1}{\sqrt{2}}\left(\left(\sqrt{2}-1\right)|\frac{1}{2}\rangle \;\;+|1-\frac{1}{2}\rangle \right)=|\frac{1}{2}\rangle$;${\sqrt{I}}^{\left(1\right)}|\frac{2}{2}\rangle =\frac{1}{\sqrt{2}}\left(-|\frac{2}{2}\rangle \;\;+|1-\frac{2}{2}\rangle \right)=\frac{1}{\sqrt{2}}\left(|\frac{0}{2}\rangle \;\;-|\frac{2}{2}\rangle \right)=\frac{1}{\sqrt{2}}\left(|0\rangle \;\;-|1\rangle \right)$.

Another important “genuine” quantum gate is the *square root of negation gate*, that can be defined as follows.

**Definition** **5.**
*The square root of negation on $H_{d}^{\left(1\right)}$*

*The square root of negation on the space $H_{d}^{\left(1\right)}$ is the linear operator ${\sqrt{\mathrm{NOT}}}^{\left(1\right)}$ such that for every element |x〉 of the canonical basis:*
$${\sqrt{\mathrm{NOT}}}^{\left(1\right)}|x\rangle =\frac{1}{2}(\left(1+ı\right)|x\rangle +\left(1-ı\right)|1-x\rangle ).$$


Clearly, the basic property of the square root of negation gate is the following:$${\sqrt{\mathrm{NOT}}}^{\left(1\right)}{\sqrt{\mathrm{NOT}}}^{\left(1\right)}|\psi \rangle ={\mathrm{NOT}}^{\left(1\right)}|\psi \rangle ,$$
for any qubit |ψ〉.

As expected, both gates ${\sqrt{I}}^{\left(1\right)}$ and ${\sqrt{\mathrm{NOT}}}^{\left(1\right)}$ can be generalized to higher-dimensional spaces:${\sqrt{I}}^{\left(n\right)}|x_{1},\ldots ,x_{n}\rangle =|x_{1},\ldots ,x_{n-1}\rangle ⊗{\sqrt{I}}^{\left(1\right)}|x_{n}\rangle .$${\sqrt{\mathrm{NOT}}}^{\left(n\right)}|x_{1},\ldots ,x_{n}\rangle =|x_{1},\ldots ,x_{n-1}\rangle ⊗{\sqrt{\mathrm{NOT}}}^{\left(1\right)}|x_{n}\rangle .$

Any unitary operator G of $H_{d}^{\left(n\right)}$ can be canonically associated to a *unitary operation*
${}^{D}G$ that transforms all density operators ρ in a reversible way, according to the following rule:$${}^{D}G\;\rho =G\;\rho \;G^{†},$$
where $G^{†}$ is the adjoint of G.

We will consider here a minimal Łukasiewicz quantum computational language $L^{Ł}$ whose alphabet contains:*atomic formulas*, including two special formulas f and t that denote the *Falsity* and the *Truth*, respectively.the following logical connectives:
the ternary *lattice-connective* ⊺ and the ternary *Łukasiewicz connective*
${⊺}_{Ł}$, corresponding to the Bertini gate combined with swap gates to bring the target to the last subsystem;the connective *square root of negation* $\sqrt{\neg }$, corresponding to the gate $\sqrt{\mathrm{NOT}}$;the *Hadamard-connective* $\sqrt{id}$, corresponding to the *Hadamard gate*.

At a syntactical level, these connectives *simulate* the behavior of the corresponding gates. While the square root of negation and the Hadamard-connective are 1-ary connectives, the other connectives are ternary connectives: if α, β are formulas and q is an atomic formula, then ${⊺}_{Ł}\left(\alpha ,\beta ,q\right)$ and ⊺(α,β,q) are formulas.

Recalling the definition of the two conjunctions And and ${\mathrm{And}}_{Ł}$, two binary conjunction-connectives ∧ and $\wedge_{Ł}$ can be defined in terms of the above connectives:$$\alpha \wedge \beta :=⊺\left(\alpha ,\beta ,f\right)\;\alpha \wedge_{Ł}\beta :={⊺}_{Ł}\left(\alpha ,\beta ,f\right).$$
where the Falsity f plays the role of a syntactical ancilla.

The *negation* ¬ can be defined in terms of the square root of negation:$$\neg \alpha :=\sqrt{\neg }\sqrt{\neg }\alpha .$$

The modal connective ⋄ can be defined by ${⊺}_{Ł}$:$$⋄\alpha :={⊺}_{Ł}\left(\alpha ,t,t\right).$$

A syntactical notion that plays an important semantic role is the concept of *atomic complexity* of a formula that is the number of occurrences of atomic formulas. For instance, At(α)=3, where $\alpha =q\wedge_{Ł}\neg q={⊺}_{Ł}\left(q,\sqrt{\neg }\sqrt{\neg }q,f\right)$ and q is an atomic formula.

For any choice of the truth-value number *d*, the number At(α) determines the *semantic space*
$H_{d}^{\alpha }$ where any piece of quantum information representing a possible meaning of α shall live. Let At(α)=n. The semantic space $H_{d}^{\alpha }$ is determined as follows:$$H_{d}^{\alpha }=H_{d}^{\left(At(\alpha )\right)}=\underset{n-times}{\underset{︸}{C^{d}⊗\ldots ⊗C^{d}}}.$$

Any formula α can be decomposed into its parts, determining a syntactical tree of α. For instance, the syntactical tree of $\alpha =\left(q\wedge \neg q\right)\wedge_{Ł}\sqrt{id}\;q={⊺}_{Ł}(⊺\left(q,\neg q,f\right),$
$\sqrt{id}\;q,f)$ is the following sequence of four *levels*, where each level is a sequence of subformulas of α:$$Level_{4}^{\alpha }=\left(q,q,f,q,f\right)$$
$$Level_{3}^{\alpha }=\left(q,\neg q,f,q,f\right)$$
$$Level_{2}^{\alpha }=\left(⊺\left(q,\neg q,f\right),\sqrt{id}\;q,f\right)$$
$$Level_{1}^{\alpha }=\left({⊺}_{Ł}\left(⊺\left(q,\neg q,f\right),\sqrt{id}\;q,f\right)\right)$$
This concept can be naturally generalized to all formulas of the language. The *bottom level* $Level_{1}^{\alpha }$ is (α). The *top level* $Level_{h}^{\alpha }$ is the sequence of atomic formulas occurring in α. Each $Level_{i+1}$ (where 1≤i<h) is obtained by dropping the *principal connective* in all molecular formulas occurring at $Level_{i}$ and by repeating all atomic formulas that occur at $Level_{i}$. The syntactical tree of any formula α gives rise a quantum circuit defined on the semantic space of α. For instance, consider again the formula
$$\alpha =\left(q\wedge \neg q\right)\wedge_{Ł}\sqrt{id}\;q={⊺}_{Ł}\left(⊺\left(q,\neg q,f\right),\sqrt{id}\;q,f\right).$$

The third level of the syntactical tree of α has been obtained from the fourth level by repeating the first occurrence of q, by negating the second occurrence of q and by repeating f, the third occurrence of q and f. The second level has been obtained from the third level by applying the lattice-connective to the three sentences occurring at the second level, by applying the square root of identity at the third occurrence of q and by repeating f. The first level has been obtained from the second level by applying the Łukasiewicz-conjunction to the three sentences occurring at the second level. Accordingly, the *gate tree* of α can be naturally identified with the following sequence of gates defined on the semantic space of α: $$(I^{\left(1\right)}⊗{\sqrt{\mathrm{NOT}}}^{\left(1\right)}{\sqrt{\mathrm{NOT}}}^{\left(1\right)}⊗I^{\left(1\right)}⊗I^{\left(1\right)}⊗I^{\left(1\right)},{\mathrm{Swap}}_{2,3}^{\left(3\right)}\;B\;{\mathrm{Swap}}_{1,3}^{\left(3\right)}\;⊗{\sqrt{I}}^{\left(1\right)}⊗I^{\left(1\right)},I^{\left(1\right)}⊗I^{\left(1\right)}⊗B).$$
Clearly, this procedure can be generalized to all formulas of the language.

As so happens in most semantic theories, the basic notion is the concept of a model that provides an interpretation for all linguistic expressions. In quantum computational languages we will handle a *holistic* form of semantics, whose models will assign a global meaning to each formula. A *d-valued holistic map* of $L^{Ł}$ is a map ${\mathrm{Hol}}_{d}$ that assigns to each level of the syntactical tree of any formula α a global interpretation, represented by a density operator living in the semantic space $H_{d}^{\alpha }$ of α. Given a formula α, any holistic map ${\mathrm{Hol}}_{d}$ determines the *contextual meaning* with respect to the context ${\mathrm{Hol}}_{d}\left(\alpha \right)$ of any occurrence of a subformula β in α. Suppose that
$$Level_{i}^{\alpha }=\left(\beta_{i_{1}},\ldots ,\beta_{i_{r}}\right).$$
The *contextual meaning* of $\beta_{i_{j}}$ with respect to the context ${\mathrm{Hol}}_{d}\left(\alpha \right)$ can be naturally defined using the notion of a *reduced state*:$${\mathrm{Hol}}_{d}^{\alpha }\left(\beta_{i_{j}}\right):=Red^{\left(j\right)}\left(\mathrm{Hol}\left(Level_{i}\left(\alpha \right)\right)\right).$$
A *d-valued holistic model* of the language $L^{Ł}$ is a *d*-valued holistic map ${\mathrm{Hol}}_{d}$ that satisfies the following conditions for any formula α.

(1)The meaning of each level is obtained by applying the corresponding gate to the meaning of the above level. Let ${(}^{D}G_{\left(h-1\right)}^{\alpha },\ldots ,{\;}^{D}G_{\left(1\right)}^{\alpha })$ be the gate tree of α and let 1≤i<h). Then,
$${\mathrm{Hol}}_{d}\left(Level_{i}^{\alpha }\right)=\;{\;}^{D}G_{i}^{\alpha }\left({\mathrm{Hol}}_{d}\left(Level_{i+1}^{\alpha }\right)\right).$$(2)${\mathrm{Hol}}_{d}$ assigns the same contextual meaning to different occurrences of the same subformula of α.(3)The contextual meanings assigned to the false sentence and to the true sentence are the Falsity and the Truth (${\mathrm{Hol}}_{d}^{\alpha }\left(f\right)=P_{|0\rangle }$
${\mathrm{Hol}}_{d}^{\alpha }\left(t\right)=P_{|1\rangle }$).

Notice that any *meaning* ${\mathrm{Hol}}_{d}\left(\alpha \right)={\mathrm{Hol}}_{d}\left(Level_{1}^{\alpha }\right)$ represents a kind of *autonomous semantic context* that is not necessarily correlated with the meanings of other sentences. In fact, the same formula may receive different contextual meanings in different contexts (${\mathrm{Hol}}_{d}^{\alpha }\left(\gamma \right)\neq {\mathrm{Hol}}_{d}^{\beta }\left(\gamma \right)$) as so happens in the case of natural languages.

We sum up some important properties of *d*-holistic models:(1)For any model ${\mathrm{Hol}}_{d}$, for any formula γ such that $\sqrt{id}\beta$ is a subformula of γ:
$${\mathrm{Hol}}_{d}^{\gamma }\left(\sqrt{id}\beta \right)=\;{\;}^{D}{\sqrt{I}}^{\left(At(\beta )\right)}{\mathrm{Hol}}_{d}^{\gamma }\left(\beta \right).$$(2)For any model ${\mathrm{Hol}}_{d}$, for any formula γ such that $\sqrt{\neg }\beta$ is a subformula of γ:
$${\mathrm{Hol}}_{d}^{\gamma }\left(\sqrt{\neg }\beta \right)=\;{\;}^{D}{\sqrt{\mathrm{NOT}}}^{\left(At(\beta )\right)}{\mathrm{Hol}}_{d}^{\gamma }\left(\beta \right).$$(3)For any model ${\mathrm{Hol}}_{d}$, for any formula γ such that ¬β is a subformula of γ:
$$p\left({\mathrm{Hol}}_{d}^{\gamma }\left(\neg \beta \right)\right)=1-p\left({\mathrm{Hol}}_{d}^{\gamma }\left(\beta \right)\right).$$
The concepts of *truth* and of *logical consequence* can be defined in the following way.

**Definition** **6.**
*Truth*
*A formula α is called* true *with respect to a model ${\mathrm{Hol}}_{d}$ iff $p({\mathrm{Hol}}_{d}\left(\alpha \right))=1$.*

**Definition** **7.**
*Logical consequence*
*A formula β is called a* logical consequence *of a formula α (α⊧β) iff for any d≥2, for any formula γ such that α and β are subformulas of γ and for any model ${\mathrm{Hol}}_{d}$,*
$$p\left({\mathrm{Hol}}_{d}^{\gamma }\left(\alpha \right)\right)\;\leq \;p\left({\mathrm{Hol}}_{d}^{\gamma }\left(\beta \right)\right).$$

We call *Łukasiewicz quantum computational logic* (**ŁQCL**) the logic that is semantically characterized by the logical consequence relation.

Clearly, the qudit-semantics includes as an important special case the qubit-semantics. We will deal with a very weak form of logic, where many important *logical arguments* such as Birkhoff and von Neumann’s quantum logic may be violated. Generally, the conjunction is
*non-idempotent*:
$$\alpha {⊭}_{\mathrm{HQCL}}\alpha \wedge \alpha .$$*non-commutative*:
$$\alpha \wedge \beta {⊭}_{\mathrm{HQCL}}\beta \wedge \alpha .$$*non-associative*:
$$\alpha \wedge \left(\beta \wedge \gamma \right){⊭}_{\mathrm{HQCL}}\left(\alpha \wedge \beta \right)\wedge \gamma .$$
Such situations can be explained by recalling the contextual behavior of quantum meanings.

Different variants of the logic HQCL have been studied and applied to investigate semantic phenomena where *holism, contextuality and ambiguity* play an important role, as so happens in the languages of art [19]. Of course this holistic semantics does not forbid *compositional situations*, where all models behave in a compositional way.

**Definition** **8.**
*Compositional model*
*A model ${\mathrm{Hol}}_{d}$ is called* compositional *iff for any formula α of the language, ${\mathrm{Hol}}_{d}$ assigns to the top level*
$$Level_{h}^{\alpha }=\left(q_{1},\ldots ,q_{r}\right)$$
*of the syntactical tree of α the following factorized state:*
$${\mathrm{Hol}}_{d}^{\alpha }\left(q_{1}\right)⊗\ldots ⊗{\mathrm{Hol}}_{d}^{\alpha }\left(q_{r}\right).$$

Any compositional model ${\mathrm{Hol}}_{d}$ assigns to each level of the syntactical tree the tensor product of the contextual meanings of the subformulas that occur at that level.

We call *compositional quantum computational logics* (CQCL) the logic characterized by the special version of the semantics where all models are compositional. In compositional semantics, one cannot recover some entangled situations. For instance, consider the sentence γ=q∧q with the following meaning $P_{\frac{1}{\sqrt{2}}(|000\rangle +\left|111\rangle \right)}$, which is an entangled pure state. The contextual meaning of the atomic formula q is $\frac{1}{2}I^{\left(1\right)}$, which is a mixed state. Clearly, we have:$$p\left(\mathrm{And}\left(\frac{1}{2}I^{\left(1\right)}⊗\frac{1}{2}I^{\left(1\right)}\right)\right)=\frac{1}{4}\neq \frac{1}{2}=p\left(P_{\frac{1}{\sqrt{2}}(|000\rangle +\left|111\rangle \right)}\right)$$
In the compositional semantics, conjunctions are always commutative and associative, but non-idempotent. Apparently, we have:$$\alpha \;\;{⊧}_{\mathrm{HQCL}}\;\;\beta \;\;\;⟹\;\;\;\alpha \;\;{⊧}_{\mathrm{CQCL}}\;\;\beta .$$
At the same time, the inverse relation does not hold.

## 3. Qubit-Semantics

It is natural to assume that the logic HQCL is formalized in the sublanguage L of $L^{Ł}$ that does not contain the Łukasiewicz connective. We will indicate by **ŁQCL*** the sublogic of **ŁQCL** formalized in the language L. Consider now two formulas α and β that belong to the language L. Clearly, we have:$$\alpha {⊧}_{{\mathrm{ŁQCL}}^{*}}\beta \;\Rightarrow \;\alpha {⊧}_{\mathrm{HQCL}}\;\beta .$$
What about the inverse implication? Recalling what happens in Łukasiewicz logics, we could expect that HQCL is stronger than ${\mathrm{ŁQCL}}^{*}$. We conjecture that this is not the case: HQCL and ${\mathrm{ŁQCL}}^{*}$ are the same logic. Apparently, quantum holism and quantum uncertainties seem to render irrelevant the use of intermediate truth-values.

The following Lemmas and Theorems are useful for proving that qubit and qutrit semantics characterize the same holistic logic.

**Lemma** **1.**
*Let ρ be a density operator of $C^{d}$ and $b=(b_{1},\ldots ,b_{d^{2}-1})$ the corresponding Bloch vector. The following conditions are satisfied:*
*(i)* 
*$p\left(\rho \right)=\frac{1-a_{3}}{2}$;*
*(ii)* 
*$p\left({\;}^{D}{\mathrm{NOT}}^{\left(1\right)}\rho \right)=\frac{1+a_{3}}{2}$;*
*(iii)* 
*$p\left({\;}^{D}{\sqrt{I}}^{\left(1\right)}\rho \right)=\frac{1-a_{1}}{2}$;*
*(iv)* 
*$p\left({\;}^{D}{\sqrt{\mathrm{NOT}}}^{\left(1\right)}\rho \right)=\frac{1-a_{2}}{2}$,*

*where $a_{1}=\sqrt{\frac{2}{d(d-1)}}\sum_{j=0}^{\lceil \frac{d}{2}\rceil -1}\left(d-1-2j\right)b_{\left(d-2-\frac{j}{2}\right)\left(j+1\right)+1}$*

$a_{2}=\sqrt{\frac{2}{d(d-1)}}\sum_{j=0}^{\lceil \frac{d}{2}\rceil -1}\left(d-1-2j\right)b_{\frac{\left(d-1)(d+2)+j(2d-5-j\right)}{2}}$

*$a_{3}=\sqrt{\frac{2}{d(d-1)}}\sum_{j=1}^{d-1}\sqrt{\frac{j(j+1)}{2}}b_{d(d-1)+j}$.*


**Proof.** Easy computation (see Appendix A). □

In terms of complex coefficients, we have:

**Lemma** **2.**
*Let $\rho =\sum_{j,k=0}^{d-1}c_{j,k}|\frac{j}{d-1}\rangle \langle \frac{k}{d-1}|$ be a density operator of $C^{d}$. The following conditions are satisfied:*
*(i)* 
*$p\left(\rho \right)=\sum_{j=1}^{d-1}\frac{j}{d-1}c_{j,j}$;*
*(ii)* 
*$p\left({\;}^{D}{\mathrm{NOT}}^{\left(1\right)}\rho \right)=\sum_{j=0}^{d-2}\frac{d-1-j}{d-1}c_{j,j}$;*
*(iii)* 
*$p\left({\;}^{D}{\sqrt{I}}^{\left(1\right)}\rho \right)=\frac{1}{2}-\sum_{j=0}^{\left\lfloor \frac{d}{2}\right\rfloor -1}\frac{d-1-2j}{d-1}\;Re\left(c_{j,d-1-j}\right)$;*
*(iv)* 
*$p\left({\;}^{D}{\sqrt{\mathrm{NOT}}}^{\left(1\right)}\rho \right)=\frac{1}{2}-\sum_{j=0}^{\left\lfloor \frac{d}{2}\right\rfloor -1}\frac{d-1-2j}{d-1}\;Im\left(c_{j,d-1-j}\right)$,*

*where $\left\lfloor \frac{d}{2}\right\rfloor$ is the integer part of $\frac{d}{2}$, Re(c) and Im(c) are the real and imaginary parts of c.*


**Proof.** Easy computation. □

**Lemma** **3.**
*Let ρ be a density operator of $C^{3}$. There exists a density operator $\tilde{\rho }$ of $C^{2}$ such that the following conditions are satisfied:*
*(i)* 
*p(ρ)=p(ρ˜);*
*(ii)* 
*pDNOT(1)ρ=pDNOT(1)ρ˜;*
*(iii)* 
*pDI(1)ρ=pDI(1)ρ˜;*
*(iv)* 
*pDNOT(1)ρ=pDNOT(1)ρ˜.*



**Proof.** Let $\rho =\sum_{j,k=0}^{2}c_{j,k}|\frac{j}{2}\rangle \langle \frac{k}{2}|$ be a density operator of $C^{3}$.Since ρ is a positive (semi)definite operator,
$$c_{0,0}\left|0\rangle \langle 0\right|+c_{2,0}\left|1\rangle \langle 0\right|+c_{0,2}\left|0\rangle \langle 1\right|+c_{2,2}\left|1\rangle \langle 1\right|$$
and
$$c_{1,1}|\frac{1}{2}\rangle \langle \frac{1}{2}|$$
are positive (semi)definite operators corresponding to principal submatrices.Consider the following Hermitian operator $\hat{\rho }$ of $C^{3}$: $\hat{\rho }=\frac{1}{2}c_{1,1}(|1\rangle \langle 1|+|0\rangle \langle 0|)+c_{0,0}\left|0\rangle \langle 0\right|+c_{2,0}\left|1\rangle \langle 0\right|+c_{0,2}\left|0\rangle \langle 1\right|+c_{2,2}\left|1\rangle \langle 1\right|$ represented by the matrix
$$\left[\begin{array}{ccc} c_{0,0}+\frac{1}{2}c_{1,1} & 0 & c_{0,2}\\ 0 & 0 & 0\\ c_{2,0} & 0 & \frac{1}{2}c_{1,1}+c_{2,2} \end{array}\right]$$
$\hat{\rho }$ is a sum of positive (semi)definite operators. Therefore, it is a positive (semi)definite operator. Clearly, $\mathrm{tr}\left(\hat{\rho }\right)=c_{0,0}+c_{1,1}+c_{2,2}=\mathrm{tr}\left(\rho \right)=1$.Consider now the operator $\tilde{\rho }$ of $C^{2}$ represented by the following principal submatrix:
$$\left[\begin{array}{cc} c_{0,0}+\frac{1}{2}c_{1,1} & c_{0,2}\\ c_{2,0} & \frac{1}{2}c_{1,1}+c_{2,2} \end{array}\right]$$
By construction, it is self-adjoint and positive (semi)definite operator with trace 1. Thus, $\tilde{\rho }$ is a density operator of $C^{2}$ and by Lemma 2,
$$p\left(\tilde{\rho }\right)=\frac{1}{2}c_{1,1}+c_{2,2}=p\left(\rho \right),$$
$$p\left({\;}^{D}{\mathrm{NOT}}^{\left(1\right)}\tilde{\rho }\right)=c_{0,0}+\frac{1}{2}c_{1,1}=p\left({\;}^{D}{\mathrm{NOT}}^{\left(1\right)}\rho \right),$$
$$p\left({\;}^{D}{\sqrt{I}}^{\left(1\right)}\tilde{\rho }\right)=\frac{1}{2}-Re\left(c_{0,2}\right)=p\left({\;}^{D}{\sqrt{I}}^{\left(1\right)}\rho \right),$$
$$p\left({\;}^{D}{\sqrt{\mathrm{NOT}}}^{\left(1\right)}\tilde{\rho }\right)=\frac{1}{2}-Im\left(c_{0,2}\right)=p\left({\;}^{D}{\sqrt{\mathrm{NOT}}}^{\left(1\right)}\rho \right).$$ □

**Lemma** **4.**
*Let ρ be a density operator of $C^{3}⊗C^{3}$. There exists a density operator $\tilde{\rho }$ of $C^{2}⊗C^{2}$ such that the following conditions are satisfied:*
*(i)* 
*$p\left({\mathrm{AND}}^{\left(1,1\right)}\left(\rho \right)\right)=p\left({\mathrm{AND}}^{\left(1,1\right)}\left(\tilde{\rho }\right)\right)$;*
*(ii)* 
*pRed[1,1](i)(ρ)=pRed[1,1](i)(ρ˜),*
*(iii)* 
*pDI(1)Red[1,1](i)(ρ)=pDI(1)Red[1,1](i)(ρ˜),*
*(iv)* 
*$p\left({\;}^{D}{\sqrt{\mathrm{NOT}}}^{\left(1\right)}Red_{\left[1,1\right]}^{\left(i\right)}\left(\rho \right)\right)=p\left({\;}^{D}{\sqrt{\mathrm{NOT}}}^{\left(1\right)}Red_{\left[1,1\right]}^{\left(i\right)}\left(\tilde{\rho }\right)\right)$, for i∈{1,2};*
*(v)* 
*if Red[1,1](1)(ρ)=Red[1,1](2)(ρ), then Red[1,1](1)(ρ˜)=Red[1,1](2)(ρ˜).*



**Proof.** Let $\rho =\sum_{x_{1},y_{1},j,k=0}^{2}c_{x_{1},j,y_{1},k}|\frac{x_{1}}{2},\frac{j}{2}\rangle \langle \frac{y_{1}}{2},\frac{k}{2}|$ be a density operator of $C^{3}⊗C^{3}$. Since ρ is a positive (semi)definite operator, $c_{x_{1},0,y_{1},0}|\frac{x_{1}}{2},0\rangle \langle \frac{y_{1}}{2},0|+c_{x_{1},2,y_{1},0}|\frac{x_{1}}{2},1\rangle \langle \frac{y_{1}}{2},0|+c_{x_{1},0,y_{1},2}|\frac{x_{1}}{2},0\rangle \langle \frac{y_{1}}{2},1|+c_{x_{1},2,y_{1},2}|\frac{x_{1}}{2},1\rangle$
$\langle \frac{y_{1}}{2},1|$ and $c_{x_{1},1,y_{1},1}|\frac{x_{1}}{2},\frac{1}{2}\rangle \langle \frac{y_{1}}{2},\frac{1}{2}|$ are positive (semi)definite operators corresponding to principal submatrices, for $x_{1},y_{1}\in \left\{0,1,2\right\}$.Consider the following operator ${\hat{\rho }}_{1}$ of $C^{3}⊗C^{3}$:
$$\begin{array}{cc} & \sum_{x_{1},y_{1}=0}^{2}c_{x_{1},0,y_{1},0}|\frac{x_{1}}{2},0\rangle \langle \frac{y_{1}}{2},0|+c_{x_{1},2,y_{1},0}|\frac{x_{1}}{2},1\rangle \langle \frac{y_{1}}{2},0|\\ & +c_{x_{1},0,y_{1},2}|\frac{x_{1}}{2},0\rangle \langle \frac{y_{1}}{2},1|+c_{x_{1},2,y_{1},2}|\frac{x_{1}}{2},1\rangle \langle \frac{y_{1}}{2},1|\\ & +\frac{1}{2}(\sum_{x_{1}=0}^{2}c_{x_{1},1,0,1}(|\frac{x_{1}}{2},0\rangle \langle 0,0|+|\frac{x_{1}}{2},1\rangle \langle 0,1|)\\ & +c_{0,1,1,1}(|0,0\rangle \langle \frac{1}{2},0|+|0,1\rangle \langle \frac{1}{2},1|)\\ & +c_{1,1,1,1}(|0,0\rangle \langle 0,0|+|1,1\rangle \langle 1,1|)\\ & +c_{2,1,1,1}(|1,0\rangle \langle \frac{1}{2},0|+|1,1\rangle \langle \frac{1}{2},1|)\\ & +\sum_{x_{1}=0}^{2}c_{x_{1},1,2,1}(|\frac{x_{1}}{2},0\rangle \langle 1,0|+|\frac{x_{1}}{2},1\rangle \langle 1,1|)) \end{array}$$
By construction, it is Hermitian and it is a sum of positive (semi)definite operators. We have: $\mathrm{tr}\left({\hat{\rho }}_{1}\right)=\sum_{x_{1},j=0}^{2}c_{x_{1},j,x_{1},j}=\mathrm{tr}\left(\rho \right)=1$. Therefore, ${\hat{\rho }}_{1}$ is a density operator.We apply a similar procedure starting from ${\hat{\rho }}_{1}=\sum_{x_{2},y_{2},j,k=0}^{2}{\hat{c}}_{j,x_{2},k,y_{2}}|\frac{j}{2},\frac{x_{2}}{2}\rangle \langle \frac{k}{2},\frac{y_{2}}{2}|$ obtaining the following density operator:
$$\begin{array}{cc} & {\hat{\rho }}_{2}=\sum_{x_{2},y_{2}=0}^{2}{\hat{c}}_{0,x_{2},0,y_{2}}|0,\frac{x_{2}}{2}\rangle \langle 0,\frac{y_{2}}{2}|+{\hat{c}}_{2,x_{2},0,y_{2}}|1,\frac{x_{2}}{2}\rangle \langle 0,\frac{y_{2}}{2}|\\ & +{\hat{c}}_{0,x_{2},2,y_{2}}|0,\frac{x_{2}}{2}\rangle \langle 1,\frac{y_{2}}{2}|+{\hat{c}}_{2,x_{2},2,y_{2}}|1,\frac{x_{2}}{2}\rangle \langle 1,\frac{y_{2}}{2}|\\ & +\frac{1}{2}{\hat{c}}_{1,x_{2},1,y_{2}}(|0,\frac{x_{2}}{2}\rangle \langle 0,\frac{y_{2}}{2}|+|1,\frac{y_{2}}{2}\rangle \langle 1,\frac{y_{2}}{2}|) \end{array}$$
represented by the matrix:
$$\left[\begin{array}{ccccccccc} c_{0,0,0,0}+\frac{1}{2}(c_{0,1,0,1} & 0 & c_{0,0,0,2}+\frac{1}{2}c_{1,0,1,2} & 0 & 0 & 0 & c_{0,0,2,0}+\frac{1}{2}c_{0,1,2,1} & 0 & c_{0,0,2,2}\\ +c_{1,0,1,0}+c_{1,1,1,1})\\ 0 & 0 & 0 & 0 & 0 & 0 & 0 & 0 & 0\\ c_{0,2,0,0}+\frac{1}{2}c_{1,2,1,0} & 0 & \frac{1}{2}c_{0,1,0,1}+c_{0,2,0,2}+\frac{1}{2}c_{1,2,1,2} & 0 & 0 & 0 & c_{0,2,2,0} & 0 & \frac{1}{2}c_{0,1,2,1}+c_{0,2,2,2}\\ 0 & 0 & 0 & 0 & 0 & 0 & 0 & 0 & 0\\ 0 & 0 & 0 & 0 & 0 & 0 & 0 & 0 & 0\\ 0 & 0 & 0 & 0 & 0 & 0 & 0 & 0 & 0\\ c_{2,0,0,0}+\frac{1}{2}c_{2,1,0,1} & 0 & c_{2,0,0,2} & 0 & 0 & 0 & \frac{1}{2}c_{1,0,1,0}+c_{2,0,2,0}+\frac{1}{2}c_{2,1,2,1} & 0 & \frac{1}{2}c_{1,0,1,2}+c_{2,0,2,2}\\ 0 & 0 & 0 & 0 & 0 & 0 & 0 & 0 & 0\\ c_{2,2,0,0} & 0 & \frac{1}{2}c_{2,1,0,1}+c_{2,2,0,2} & 0 & 0 & 0 & \frac{1}{2}c_{1,2,1,0}+c_{2,2,2,0} & 0 & \frac{1}{2}(c_{1,1,1,1}+c_{1,2,1,2}\\ & & & & & & & & +c_{2,1,2,1})+c_{2,2,2,2} \end{array}\right]$$One can naturally define the density operator $\tilde{\rho }$ of $C^{2}⊗C^{2}$ corresponding to the principal submatrix:
$$\left[\begin{array}{cccc} c_{0,0,0,0}+\frac{1}{2}(c_{0,1,0,1} & c_{0,0,0,2}+\frac{1}{2}c_{1,0,1,2} & c_{0,0,2,0}+\frac{1}{2}c_{0,1,2,1} & c_{0,0,2,2}\\ +c_{1,0,1,0}+c_{1,1,1,1})\\ c_{0,2,0,0}+\frac{1}{2}c_{1,2,1,0} & \frac{1}{2}c_{0,1,0,1}+c_{0,2,0,2}+\frac{1}{2}c_{1,2,1,2} & c_{0,2,2,0} & \frac{1}{2}c_{0,1,2,1}+c_{0,2,2,2}\\ c_{2,0,0,0}+\frac{1}{2}c_{2,1,0,1} & c_{2,0,0,2} & \frac{1}{2}c_{1,0,1,0}+c_{2,0,2,0}+\frac{1}{2}c_{2,1,2,1} & \frac{1}{2}c_{1,0,1,2}+c_{2,0,2,2}\\ c_{2,2,0,0} & \frac{1}{2}c_{2,1,0,1}+c_{2,2,0,2} & \frac{1}{2}c_{1,2,1,0}+c_{2,2,2,0} & \frac{1}{2}(c_{1,1,1,1}+c_{1,2,1,2}\\ & & & +c_{2,1,2,1})+c_{2,2,2,2} \end{array}\right]$$Thus, by easy computation, the following conditions are satisfied:
$p\left({\mathrm{AND}}^{\left(1,1\right)}\left(\rho \right)\right)=\frac{1}{2}\left(c_{1,1,1,1}+c_{1,2,1,2}+c_{2,1,2,1}\right)+c_{2,2,2,2}=p\left({\mathrm{AND}}^{\left(1,1\right)}\left(\tilde{\rho }\right)\right)$;$p\left(Red_{\left[1,1\right]}^{\left(1\right)}\left(\rho \right)\right)=\sum_{x_{2}=0}^{2}\sum_{j=1}^{2}\frac{j}{2}c_{j,x_{2},j,x_{2}}=p\left(Red_{\left[1,1\right]}^{\left(1\right)}\left(\tilde{\rho }\right)\right)$;$p\left({\;}^{D}{\sqrt{I}}^{\left(1\right)}Red_{\left[1,1\right]}^{\left(1\right)}\left(\rho \right)\right)=\frac{1}{2}-\sum_{x_{2}=0}^{2}Re\left(c_{0,x_{2},2,x_{2}}\right)=p\left({\;}^{D}{\sqrt{I}}^{\left(1\right)}Red_{\left[1,1\right]}^{\left(1\right)}\left(\tilde{\rho }\right)\right)$;$p\left({\;}^{D}{\sqrt{\mathrm{NOT}}}^{\left(1\right)}Red_{\left[1,1\right]}^{\left(1\right)}\left(\rho \right)\right)=\frac{1}{2}-\sum_{x_{2}=0}^{2}Im\left(c_{0,x_{2},2,x_{2}}\right)=p\left({\;}^{D}{\sqrt{\mathrm{NOT}}}^{\left(1\right)}Red_{\left[1,1\right]}^{\left(1\right)}\left(\tilde{\rho }\right)\right)$;$p\left(Red_{\left[1,1\right]}^{\left(2\right)}\left(\rho \right)\right)=\sum_{x_{1}=0}^{2}\sum_{j=1}^{2}\frac{j}{2}c_{x_{1},j,x_{1},j}=p\left(Red_{\left[1,1\right]}^{\left(2\right)}\left(\tilde{\rho }\right)\right)$;$p\left({\;}^{D}{\sqrt{I}}^{\left(1\right)}Red_{\left[1,1\right]}^{\left(2\right)}\left(\rho \right)\right)=\frac{1}{2}-\sum_{x_{1}=0}^{2}Re\left(c_{x_{1},0,x_{1},2}\right)=p\left({\;}^{D}{\sqrt{I}}^{\left(1\right)}Red_{\left[1,1\right]}^{\left(2\right)}\left(\tilde{\rho }\right)\right)$;$p\left({\;}^{D}{\sqrt{\mathrm{NOT}}}^{\left(1\right)}Red_{\left[1,1\right]}^{\left(2\right)}\left(\rho \right)\right)=\frac{1}{2}-\sum_{x_{1}=0}^{2}Im\left(c_{x_{1},0,x_{1},2}\right)=p\left({\;}^{D}{\sqrt{\mathrm{NOT}}}^{\left(1\right)}Red_{\left[1,1\right]}^{\left(2\right)}\left(\tilde{\rho }\right)\right)$.
In particular, if $Red_{\left[1,1\right]}^{\left(1\right)}\left(\rho \right)=Red_{\left[1,1\right]}^{\left(2\right)}\left(\rho \right)$, we have:
$$\sum_{x_{2}=0}^{2}Re\left(c_{0,x_{2},2,x_{2}}\right)=\sum_{x_{1}=0}^{2}Re\left(c_{x_{1},0,x_{1},2}\right);$$
$$\sum_{x_{2}=0}^{2}Im\left(c_{0,x_{2},2,x_{2}}\right)=\sum_{x_{1}=0}^{2}Im\left(c_{x_{1},0,x_{1},2}\right);$$
$$\sum_{x_{2}=0}^{2}\sum_{j=1}^{2}\frac{j}{2}c_{j,x_{2},j,x_{2}}=\sum_{x_{1}=0}^{2}\sum_{j=1}^{2}\frac{j}{2}c_{x_{1},j,x_{1},j}.$$
Consequently, $Red_{\left[1,1\right]}^{\left(1\right)}\left(\tilde{\rho }\right)=Red_{\left[1,1\right]}^{\left(2\right)}\left(\tilde{\rho }\right)$. □

**Lemma** **5.**
*Let ρ be a density operator of $C^{3}⊗C^{3}$. There exists a density operator $\tilde{\rho }$ of $D(H_{2}^{\left(2\right)})$ such that the following conditions are satisfied:*
*(i)* 
*$p\left({\mathrm{LAND}}^{\left(1,1\right)}\left(\rho \right)\right)=p\left({\mathrm{AND}}^{\left(1,1\right)}\left(\tilde{\rho }\right)\right)$;*
*(ii)* 
*pRed[1,1](i)(ρ)=pRed[1,1](i)(ρ˜),*
*(iii)* 
*pDI(1)Red[1,1](i)(ρ)=pDI(1)Red[1,1](i)(ρ˜),*
*(iv)* 
*$p\left({\;}^{D}{\sqrt{\mathrm{NOT}}}^{\left(1\right)}Red_{\left[1,1\right]}^{\left(i\right)}\left(\rho \right)\right)=p\left({\;}^{D}{\sqrt{\mathrm{NOT}}}^{\left(1\right)}Red_{\left[1,1\right]}^{\left(i\right)}\left(\tilde{\rho }\right)\right)$,*
*for i∈{1,2};*
*(v)* 
*if $Red_{\left[1,1\right]}^{\left(1\right)}\left(\rho \right)=Red_{\left[1,1\right]}^{\left(2\right)}\left(\rho \right)$, then $Red_{\left[1,1\right]}^{\left(1\right)}\left(\tilde{\rho }\right)=Red_{\left[1,1\right]}^{\left(2\right)}\left(\tilde{\rho }\right)$.*



**Proof.** Let $\rho =\sum_{x_{1},y_{1},j,k=0}^{2}c_{x_{1},j,y_{1},k}|\frac{x_{1}}{2},\frac{j}{2}\rangle \langle \frac{y_{1}}{2},\frac{k}{2}|$ be a density operator of $C^{3}⊗C^{3}$. Similar to Lemma 4, one can define the density operator $\tilde{\rho }$ of $C^{2}⊗C^{2}$ corresponding to the principal submatrix:
$$\left[\begin{array}{cccc} c_{0,0,0,0}+\frac{1}{2}(c_{0,1,0,1} & c_{0,0,0,2}+\frac{1}{2}c_{1,0,1,2} & c_{0,0,2,0}+\frac{1}{2}c_{0,1,2,1} & c_{0,0,2,2}\\ +c_{1,0,1,0})\\ c_{0,2,0,0}+\frac{1}{2}c_{1,2,1,0} & \frac{1}{2}\left(c_{0,1,0,1}+c_{1,1,1,1}\right) & c_{0,2,2,0} & \frac{1}{2}c_{0,1,2,1}+c_{0,2,2,2}\\ & +c_{0,2,0,2}+\frac{1}{2}c_{1,2,1,2}\\ c_{2,0,0,0}+\frac{1}{2}c_{2,1,0,1} & c_{2,0,0,2} & \frac{1}{2}\left(c_{1,0,1,0}+c_{1,1,1,1}\right) & \frac{1}{2}c_{1,0,1,2}+c_{2,0,2,2}\\ & & +c_{2,0,2,0}+\frac{1}{2}c_{2,1,2,1}\\ c_{2,2,0,0} & \frac{1}{2}c_{2,1,0,1}+c_{2,2,0,2} & \frac{1}{2}c_{1,2,1,0}+c_{2,2,2,0} & \frac{1}{2}(c_{1,2,1,2}\\ & & & +c_{2,1,2,1})+c_{2,2,2,2} \end{array}\right]$$Thus, the following conditions are satisfied:
$p\left({\mathrm{LAND}}^{\left(1,1\right)}\left(\rho \right)\right)=\frac{1}{2}\left(c_{1,2,1,2}+c_{2,1,2,1}\right)+c_{2,2,2,2}=p\left({\mathrm{AND}}^{\left(1,1\right)}\left(\tilde{\rho }\right)\right)$;$p\left(Red_{\left[1,1\right]}^{\left(1\right)}\left(\rho \right)\right)=\sum_{x_{2}=0}^{2}\sum_{j=1}^{2}\frac{j}{2}c_{j,x_{2},j,x_{2}}=p\left(Red_{\left[1,1\right]}^{\left(1\right)}\left(\tilde{\rho }\right)\right)$;$p\left({\;}^{D}{\sqrt{I}}^{\left(1\right)}Red_{\left[1,1\right]}^{\left(1\right)}\left(\rho \right)\right)=\frac{1}{2}-\sum_{x_{2}=0}^{2}Re\left(c_{0,x_{2},2,x_{2}}\right)=p\left({\;}^{D}{\sqrt{I}}^{\left(1\right)}Red_{\left[1,1\right]}^{\left(1\right)}\left(\tilde{\rho }\right)\right)$;$p\left({\;}^{D}{\sqrt{\mathrm{NOT}}}^{\left(1\right)}Red_{\left[1,1\right]}^{\left(1\right)}\left(\rho \right)\right)=\frac{1}{2}-\sum_{x_{2}=0}^{2}Im\left(c_{0,x_{2},2,x_{2}}\right)=p\left({\;}^{D}{\sqrt{\mathrm{NOT}}}^{\left(1\right)}Red_{\left[1,1\right]}^{\left(1\right)}\left(\tilde{\rho }\right)\right)$;$p\left(Red_{\left[1,1\right]}^{\left(2\right)}\left(\rho \right)\right)=\sum_{x_{1}=0}^{2}\sum_{j=1}^{2}\frac{j}{2}c_{x_{1},j,x_{1},j}=p\left(Red_{\left[1,1\right]}^{\left(2\right)}\left(\tilde{\rho }\right)\right)$;$p\left({\;}^{D}{\sqrt{I}}^{\left(1\right)}Red_{\left[1,1\right]}^{\left(2\right)}\left(\rho \right)\right)=\frac{1}{2}-\sum_{x_{1}=0}^{2}Re\left(c_{x_{1},0,x_{1},2}\right)=p\left({\;}^{D}{\sqrt{I}}^{\left(1\right)}Red_{\left[1,1\right]}^{\left(2\right)}\left(\tilde{\rho }\right)\right)$;$p\left({\;}^{D}{\sqrt{\mathrm{NOT}}}^{\left(1\right)}Red_{\left[1,1\right]}^{\left(2\right)}\left(\rho \right)\right)=\frac{1}{2}-\sum_{x_{1}=0}^{2}Im\left(c_{x_{1},0,x_{1},2}\right)=p\left({\;}^{D}{\sqrt{\mathrm{NOT}}}^{\left(1\right)}Red_{\left[1,1\right]}^{\left(2\right)}\left(\tilde{\rho }\right)\right)$.
In particular, if $Red_{\left[1,1\right]}^{\left(1\right)}\left(\rho \right)=Red_{\left[1,1\right]}^{\left(2\right)}\left(\rho \right)$, we have:
$$\sum_{x_{2}=0}^{2}Re\left(c_{0,x_{2},2,x_{2}}\right)=\sum_{x_{1}=0}^{2}Re\left(c_{x_{1},0,x_{1},2}\right);$$
$$\sum_{x_{2}=0}^{2}Im\left(c_{0,x_{2},2,x_{2}}\right)=\sum_{x_{1}=0}^{2}Im\left(c_{x_{1},0,x_{1},2}\right);$$
$$\sum_{x_{2}=0}^{2}\sum_{j=1}^{2}\frac{j}{2}c_{j,x_{2},j,x_{2}}=\sum_{x_{1}=0}^{2}\sum_{j=1}^{2}\frac{j}{2}c_{x_{1},j,x_{1},j}.$$
Consequently, $Red_{\left[1,1\right]}^{\left(1\right)}\left(\tilde{\rho }\right)=Red_{\left[1,1\right]}^{\left(2\right)}\left(\tilde{\rho }\right)$. □

The following Lemma shows that the reduced state with respect to the last subsystem of a given density operator ρ plays an important probabilistic role.

**Lemma** **6.**
*Let $\rho \in D(H_{d}^{\left(n\right)})$. We have:*
$$p\left(\rho \right)=p(Red_{\left[n-1,1\right]}^{\left(2\right)}\left(\rho \right)),$$
*where n>1.*


**Proof.** By definition of the probability function p. □

**Theorem** **1.**
*Consider a formula γ=¬q and a holistic model ${\mathrm{Hol}}_{3}$ of L. There exists a holistic model $\mathrm{Ho}l_{2}$ of L such that for any subformula β of γ,*
$$p\left({\mathrm{Hol}}_{2}^{\gamma }\left(\beta \right)\right)=p\left({\mathrm{Hol}}_{3}^{\gamma }\left(\beta \right)\right).$$


**Proof.** Consider the syntactical tree of γ:
$$Level_{2}^{\gamma }=\left(q\right)$$
$$Level_{1}^{\gamma }=\left(\neg \;q\right)$$By definition of a holistic model, we have:
$${\mathrm{Hol}}_{3}^{\gamma }\left(\gamma \right)={\mathrm{Hol}}_{3}\left(\neg \;q\right)=\rho^{\left[1\right]}\in D\left(C^{3}\right),$$
$${\mathrm{Hol}}_{3}^{\gamma }\left(q\right)=\rho^{\left[2\right]}\in D\left(C^{3}\right).$$By Lemma 3, there exists a density operator $\tilde{\rho }$ of $C^{2}$ such that:
$$p\left(\tilde{\rho }\right)=p\left(\rho^{\left[2\right]}\right),$$
$$p\left({\;}^{D}{\mathrm{NOT}}^{\left(1\right)}\left(\tilde{\rho }\right)\right)=p\left(\rho^{\left[1\right]}\right).$$Thus, for each density operator there exists a corresponding density operator in the “two-valued semantic world”:
$${\mathrm{Hol}}_{2}^{\gamma }\left(\gamma \right)={\;}^{D}{\mathrm{NOT}}^{\left(1\right)}\left(\tilde{\rho }\right),$$
$${\mathrm{Hol}}_{2}^{\gamma }\left(q\right)=\tilde{\rho }.$$
Consequently, there exists a model ${\mathrm{Hol}}_{2}$ of γ such that for any subformula β of γ:
$$p\left({\mathrm{Hol}}_{2}^{\gamma }\left(\beta \right)\right)=p\left({\mathrm{Hol}}_{3}^{\gamma }\left(\beta \right)\right).$$ □

**Theorem** **2.**
*Consider a formula $\gamma =\sqrt{id}\;q$ and a holistic model ${\mathrm{Hol}}_{3}$ of L. There exists a holistic model $\mathrm{Ho}l_{2}$ of L such that for any subformula β of γ,*
$$p\left({\mathrm{Hol}}_{2}^{\gamma }\left(\beta \right)\right)=p\left({\mathrm{Hol}}_{3}^{\gamma }\left(\beta \right)\right).$$


**Proof.** Similar to Theorem 1. □

**Theorem** **3.**
*Consider a formula $\gamma =\sqrt{\neg }\;q$ and a holistic model ${\mathrm{Hol}}_{3}$ of L. There exists a holistic model $\mathrm{Ho}l_{2}$ of L such that for any subformula β of γ,*
$$p\left({\mathrm{Hol}}_{2}^{\gamma }\left(\beta \right)\right)=p\left({\mathrm{Hol}}_{3}^{\gamma }\left(\beta \right)\right).$$


**Proof.** Similar to Theorem 1. □

**Theorem** **4.**
*Consider a formula $\gamma =q_{1}\wedge q_{2}$ and a holistic model ${\mathrm{Hol}}_{3}$ of L. There exists a holistic model $\mathrm{Ho}l_{2}$ of L such that for any subformula β of γ,*
$$p\left({\mathrm{Hol}}_{2}^{\gamma }\left(\beta \right)\right)=p\left({\mathrm{Hol}}_{3}^{\gamma }\left(\beta \right)\right).$$


**Proof.** Consider the syntactical tree of γ:
$$Level_{2}^{\gamma }=\left(q_{1},q_{2},f\right)$$
$$Level_{1}^{\gamma }=\left(⊺\left(q_{1},q_{2},f\right)\right)$$By definition of holistic model, we have:
$$\rho^{\left[2\right]},\rho^{\left[1\right]}\in D\left(C^{3}⊗C^{3}⊗C^{3}\right),$$
where $\rho^{\left[1\right]}={\mathrm{Hol}}_{3}\left(Level_{1}^{\gamma }\right)$ and $\rho^{\left[2\right]}={\mathrm{Hol}}_{3}\left(Level_{2}^{\gamma }\right)$.${\mathrm{Hol}}_{3}$ determines the contextual meanings of the 4 subformulas of γ:
$${\mathrm{Hol}}_{3}^{\gamma }\left(\gamma \right),{\mathrm{Hol}}_{3}^{\gamma }\left(q_{1}\right),{\mathrm{Hol}}_{3}^{\gamma }\left(q_{2}\right),{\mathrm{Hol}}_{3}^{\gamma }\left(f\right).$$We have:
$${\mathrm{Hol}}_{3}^{\gamma }\left(\gamma \right)={\mathrm{Hol}}_{3}\left(\gamma \right)=\rho^{\left[1\right]},$$
$${\mathrm{Hol}}_{3}^{\gamma }\left(q_{1}\right)=\rho_{1}^{\left[2\right]}\in D\left(C^{3}\right),$$
$${\mathrm{Hol}}_{3}^{\gamma }\left(q_{2}\right)=\rho_{2}^{\left[2\right]}\in D\left(C^{3}\right),$$
$${\mathrm{Hol}}_{3}^{\gamma }\left(f\right)=Red_{\left[1,1,1\right]}^{\left(3\right)}\left(\rho^{\left[2\right]}\right)=P_{|0\rangle }\in D\left(C^{3}\right),$$
where $\rho_{1}^{\left[2\right]}=Red_{\left[1,1,1\right]}^{\left(1\right)}\left(\rho^{\left[2\right]}\right)$, $\rho_{2}^{\left[2\right]}=Red_{\left[1,1,1\right]}^{\left(2\right)}\left(\rho^{\left[2\right]}\right)$, $P_{|0\rangle }$ represents the *Falsity*.The contextual meanings of the subformulas of γ with respect to the model ${\mathrm{Hol}}_{3}$ determine in a natural way a special configuration that we call *the semantic tree of γ determined by the model ${\mathrm{Hol}}_{3}$*:
$$SemLevel_{2}^{\gamma }=\left({\mathrm{Hol}}_{3}^{\gamma }\left(q_{1}\right),{\mathrm{Hol}}_{3}^{\gamma }\left(q_{2}\right),{\mathrm{Hol}}_{3}^{\gamma }\left(f\right)\right)$$
$$SemLevel_{1}^{\gamma }=\left({\mathrm{Hol}}_{3}^{\gamma }\left(⊺\left(q_{1},q_{2},f\right)\right)\right)$$The 4 density operators $\rho_{1}^{\left[2\right]}$, $\rho_{2}^{\left[2\right]}$, $P_{|0\rangle }$, $\rho^{\left[1\right]}$ belong to the “three-valued semantic world”. Each of them has a well determined probability value:
$$p\left(\rho_{1}^{\left[2\right]}\right),p\left(\rho_{2}^{\left[2\right]}\right),p\left(P_{|0\rangle }\right),p\left(\rho^{\left[1\right]}\right).$$Clearly, $P_{|0\rangle }$ of $C^{2}$ keep the same probability $p(P_{|0\rangle })=0$.Consider the density operator $Red_{\left[2,1\right]}^{\left(2\right)}\left(\rho^{\left[2\right]}\right)$ of $C^{3}⊗C^{3}$.By Lemma 4, there exists a density operator $\tilde{\rho }$ of $C^{2}⊗C^{2}$ such that:
$$p\left(Red_{\left[1,1\right]}^{\left(1\right)}\left(\tilde{\rho }\right)\right)=p\left(\rho_{1}^{\left[2\right]}\right),$$
$$p\left(Red_{\left[1,1\right]}^{\left(2\right)}\left(\tilde{\rho }\right)\right)=p\left(\rho_{2}^{\left[2\right]}\right),$$
$$p\left({\mathrm{AND}}^{\left(1,1\right)}\left(\tilde{\rho }\right)\right)=p\left(\rho^{\left[1\right]}\right).$$Thus, for each density operator there exists a corresponding density operator in the “two-valued semantic world”:
$${\mathrm{Hol}}_{2}^{\gamma }\left(\gamma \right)={\mathrm{AND}}^{\left(1,1\right)}\left(\tilde{\rho }\right),$$
$${\mathrm{Hol}}_{2}^{\gamma }\left(q_{1}\right)=Red_{\left[1,1\right]}^{\left(1\right)}\left(\tilde{\rho }\right),$$
$${\mathrm{Hol}}_{2}^{\gamma }\left(q_{2}\right)=Red_{\left[1,1\right]}^{\left(2\right)}\left(\tilde{\rho }\right),$$
$${\mathrm{Hol}}_{2}^{\gamma }\left(f\right)=P_{|0\rangle }.$$
In particular, if $\rho_{1}^{\left[2\right]}=\rho_{2}^{\left[2\right]}$, then, by Lemma 4, $Red_{\left[1,1\right]}^{\left(1\right)}\left(\tilde{\rho }\right)=Red_{\left[1,1\right]}^{\left(2\right)}\left(\tilde{\rho }\right)$. ${\mathrm{Hol}}_{2}$ is a model of γ since the normality-condition is satisfied. Consequently, there exists a model ${\mathrm{Hol}}_{2}$ of γ such that for any subformula β of γ:
$$p\left({\mathrm{Hol}}_{2}^{\gamma }\left(\beta \right)\right)=p\left({\mathrm{Hol}}_{3}^{\gamma }\left(\beta \right)\right).$$ □

**Theorem** **5.**
*Consider a formula $\gamma =q_{1}\wedge_{Ł}q_{2}$ and a holistic model ${\mathrm{Hol}}_{3}$ of L. There exists a holistic model $\mathrm{Ho}l_{2}$ of L such that for any subformula β of γ,*
$$p\left({\mathrm{Hol}}_{2}^{\gamma }\left(\beta \right)\right)=p\left({\mathrm{Hol}}_{3}^{\gamma }\left(\beta \right)\right).$$


**Proof.** Similar to Theorem 4. Consider a holistic model ${\mathrm{Hol}}_{3}$ of L. ${\mathrm{Hol}}_{3}$ determines the contextual meanings of the 4 subformulas of γ:
$${\mathrm{Hol}}_{3}^{\gamma }\left(\gamma \right)={\mathrm{Hol}}_{3}\left(\gamma \right)=\rho^{\left[1\right]}\in D\left(C^{3}⊗C^{3}⊗C^{3}\right),$$
$${\mathrm{Hol}}_{3}^{\gamma }\left(q_{1}\right)=\rho_{1}^{\left[2\right]}\in D\left(C^{3}\right),$$
$${\mathrm{Hol}}_{3}^{\gamma }\left(q_{2}\right)=\rho_{2}^{\left[2\right]}\in D\left(C^{3}\right),$$
$${\mathrm{Hol}}_{3}^{\gamma }\left(f\right)=P_{|0\rangle }\in D\left(C^{3}\right).$$
Consider the density operator $Red_{\left[2,1\right]}^{\left(2\right)}\left(\rho^{\left[2\right]}\right)$ of $C^{3}⊗C^{3}$.By Lemma 5 there exists a density operator $\tilde{\rho }$ of $C^{2}⊗C^{2}$ such that:
$$p\left(Red_{\left[1,1\right]}^{\left(1\right)}\left(\tilde{\rho }\right)\right)=p\left(\rho_{1}^{\left[2\right]}\right),$$
$$p\left(Red_{\left[1,1\right]}^{\left(2\right)}\left(\tilde{\rho }\right)\right)=p\left(\rho_{2}^{\left[2\right]}\right),$$
$$p\left({\mathrm{AND}}_{Ł}^{\left(1,1\right)}\left(\tilde{\rho }\right)\right)=p\left(\rho^{\left[1\right]}\right).$$Thus, there exists a model ${\mathrm{Hol}}_{2}$ of γ that assigns the corresponding density operator in the “two-valued semantic world”:
$${\mathrm{Hol}}_{2}^{\gamma }\left(\gamma \right)=\;{\mathrm{AND}}_{Ł}^{\left(1,1\right)}\left(\tilde{\rho }\right),$$
$${\mathrm{Hol}}_{2}^{\gamma }\left(q_{1}\right)=Red_{\left[1,1\right]}^{\left(1\right)}\left(\tilde{\rho }\right),$$
$${\mathrm{Hol}}_{2}^{\gamma }\left(q_{2}\right)=Red_{\left[1,1\right]}^{\left(2\right)}\left(\tilde{\rho }\right),$$
$${\mathrm{Hol}}_{2}^{\gamma }\left(f\right)=P_{|0\rangle }.$$
Consequently, there exists a model ${\mathrm{Hol}}_{2}$ of γ such that for any subformula β of γ:
$$p\left({\mathrm{Hol}}_{2}^{\gamma }\left(\beta \right)\right)=p\left({\mathrm{Hol}}_{3}^{\gamma }\left(\beta \right)\right).$$ □

**Corollary** **1.**
*Consider a formula γ with height of the syntactical tree h≤2 and a holistic model ${\mathrm{Hol}}_{3}$ of L. There exists a holistic model ${\mathrm{Hol}}_{2}$ of L such that for any subformula β of γ,*
$$p\left({\mathrm{Hol}}_{2}^{\gamma }\left(\beta \right)\right)=p\left({\mathrm{Hol}}_{3}^{\gamma }\left(\beta \right)\right).$$


**Proof.** Consider a formula γ.Let h=1 be the height of the syntactical tree of γ. Clearly, ${\mathrm{Hol}}_{3}^{\gamma }\left(\gamma \right)\in D\left(C^{3}\right)$. By Lemma 3, there exists a density operator $\tilde{\rho }$ of $C^{2}$ such that $p\left(\tilde{\rho }\right)=p\left({\mathrm{Hol}}_{3}^{\gamma }\left(\gamma \right)\right)$. Thus, there exists a holistic model ${\mathrm{Hol}}_{2}$ of L such that ${\mathrm{Hol}}_{2}^{\gamma }\left(\gamma \right))=\tilde{\rho }$ and $p\left({\mathrm{Hol}}_{2}^{\gamma }\left(\gamma \right)\right)=p\left({\mathrm{Hol}}_{3}^{\gamma }\left(\gamma \right)\right).$Let h=2. Then, γ=¬q or $\gamma =\sqrt{id}q$ or $\gamma =\sqrt{\neg }q$ or $\gamma =q_{1}\wedge q_{2}$ or $\gamma =q_{1}\wedge_{Ł}q_{2}$. By Theorems 1–5, there exists a holistic model ${\mathrm{Hol}}_{2}$ of L such that for any subformula β of γ, $p\left({\mathrm{Hol}}_{2}^{\gamma }\left(\beta \right)\right)=p\left({\mathrm{Hol}}_{3}^{\gamma }\left(\beta \right)\right).$ □

In a future paper we will extend these results by considering any formula γ in the qudit-semantics and we will prove that qubit and qudit semantics characterize the same holistic logic. The following Lemma is an extension of Lemma 3 and plays a fundamental role.

**Lemma** **7.**
*Let ρ be a density operator of $C^{d}$. There exists a density operator $\tilde{\rho }$ of $C^{2}$ such that the following conditions are satisfied:*
*(i)* 
*p(ρ)=p(ρ˜);*
*(ii)* 
*pDNOT(1)ρ=pDNOT(1)ρ˜;*
*(iii)* 
*pDI(1)ρ=pDI(1)ρ˜;*
*(iv)* 
*pDNOT(1)ρ=pDNOT(1)ρ˜.*



**Proof.** Let $\rho =\sum_{j,k=0}^{d-1}c_{j,k}|\frac{j}{d-1}\rangle \langle \frac{k}{d-1}|$ be a density operator of $C^{d}$.$c_{j,j}|\frac{j}{d-1}\rangle \langle \frac{j}{d-1}|+c_{d-1-j,j}|\frac{d-1-j}{d-1}\rangle \langle \frac{j}{d-1}|+c_{j,d-1-j}|\frac{j}{d-1}\rangle \langle \frac{d-1-j}{d-1}|+c_{d-1-j,d-1-j}|\frac{d-1-j}{d-1}\rangle$$\langle \frac{d-1-j}{d-1}|$ are positive (semi)definite operators corresponding to principal submatrices, for $j\in \{0,\ldots ,\left\lfloor \frac{d-2}{2}\right\rfloor \}$. Thus,
$$c_{j,j}\left|1\rangle \langle 1\right|+c_{d-1-j,j}\left|1\rangle \langle 0\right|+c_{j,d-1-j}\left|0\rangle \langle 1\right|+c_{d-1-j,d-1-j}\left|0\rangle \langle 0\right|$$
are positive (semi)definite operators.Similar to Lemma 3, consider the following Hermitian operator $\hat{\rho }$ of $C^{d}$:
$$\hat{\rho }=\left\{\begin{array}{cc} \frac{1}{2}c_{\frac{d-1}{2},\frac{d-1}{2}}(|1\rangle \langle 1|+|0\rangle \langle 0|)+\\ \sum_{j=0}^{\frac{d-3}{2}}\frac{j}{d-1}(c_{j,j}|1\rangle \langle 1|-c_{d-1-j,j}\left|1\rangle \langle 0\right|-c_{j,d-1-j}\left|0\rangle \langle 1\right|+c_{d-1-j,d-1-j}|0\rangle \langle 0|)\\ +\frac{d-1-j}{d-1}(c_{j,j}|0\rangle \langle 0|+c_{d-1-j,j}\left|1\rangle \langle 0\right|+c_{j,d-1-j}\left|0\rangle \langle 1\right|+c_{d-1-j,d-1-j}|1\rangle \langle 1|), & \mathrm{if}\;d\;\mathrm{is}\;\mathrm{odd};\\ \sum_{j=0}^{\frac{d-2}{2}}\frac{j}{d-1}(c_{j,j}|1\rangle \langle 1|-c_{d-1-j,j}\left|1\rangle \langle 0\right|-c_{j,d-1-j}\left|0\rangle \langle 1\right|+c_{d-1-j,d-1-j}|0\rangle \langle 0|)\\ +\frac{d-1-j}{d-1}(c_{j,j}|0\rangle \langle 0|+c_{d-1-j,j}\left|1\rangle \langle 0\right|+c_{j,d-1-j}\left|0\rangle \langle 1\right|+c_{d-1-j,d-1-j}|1\rangle \langle 1|), & \mathrm{otherwise}. \end{array}\right.$$
represented by the following matrix:
$$\left[\begin{array}{ccccc} \frac{1+b_{3}}{2} & 0 & \cdots & 0 & \frac{b_{1}-ib_{2}}{2}\\ 0 & 0 & \cdots & 0 & 0\\ \vdots & \vdots & \vdots & \vdots & \vdots \\ 0 & 0 & \cdots & 0 & 0\\ \frac{b_{1}+ib_{2}}{2} & 0 & \cdots & 0 & \frac{1-b_{3}}{2} \end{array}\right]$$
with $b_{1}=2\sum_{j=0}^{\left\lfloor \frac{d}{2}\right\rfloor -1}\frac{d-1-2j}{d-1}\;Re\left(c_{j,d-1-j}\right)$, $b_{2}=2\sum_{j=0}^{\left\lfloor \frac{d}{2}\right\rfloor -1}\frac{d-1-2j}{d-1}\;Im\left(c_{j,d-1-j}\right)$, $b_{3}=1-2\sum_{j=1}^{d-1}\frac{j}{d-1}c_{j,j}$.$\hat{\rho }$ is a sum of positive (semi)definite operators. Therefore, it is a positive (semi)definite operator. Clearly, $\mathrm{tr}\left(\hat{\rho }\right)=\sum_{j=0}^{d-1}c_{j,j}=\mathrm{tr}\left(\rho \right)=1$.Consider now the operator $\tilde{\rho }$ of $C^{2}$ represented by the following principal submatrix:
$$\left[\begin{array}{cc} \frac{1+b_{3}}{2} & \frac{b_{1}-ib_{2}}{2}\\ \frac{b_{1}+ib_{2}}{2} & \frac{1-b_{3}}{2} \end{array}\right]$$
By construction, it is self-adjoint and positive (semi)definite operator with trace 1. Thus, $\tilde{\rho }$ is a density operator of $C^{2}$ and by Lemma 2,
$$p\left(\rho \right)=\sum_{j=1}^{d-1}\frac{j}{d-1}c_{j,j}=\frac{1-b_{3}}{2}=p\left(\tilde{\rho }\right),$$
$$p\left({\;}^{D}{\mathrm{NOT}}^{\left(1\right)}\rho \right)=\sum_{j=1}^{d-1}\frac{d-1-j}{d-1}c_{j,j}=\frac{1+b_{3}}{2}=p\left({\;}^{D}{\mathrm{NOT}}^{\left(1\right)}\tilde{\rho }\right),$$
$$p\left({\;}^{D}{\sqrt{I}}^{\left(1\right)}\rho \right)=\frac{1}{2}-\sum_{j=0}^{\left\lfloor \frac{d}{2}\right\rfloor -1}\frac{d-1-2j}{d-1}\;Re\left(c_{j,d-1-j}\right)=\frac{1-b_{1}}{2}=p\left({\;}^{D}{\sqrt{I}}^{\left(1\right)}\tilde{\rho }\right),$$
$$p\left({\;}^{D}{\sqrt{\mathrm{NOT}}}^{\left(1\right)}\rho \right)=\frac{1}{2}-\sum_{j=0}^{\left\lfloor \frac{d}{2}\right\rfloor -1}\frac{d-1-2j}{d-1}\;Im\left(c_{j,d-1-j}\right)=\frac{1-b_{2}}{2}=p\left({\;}^{D}{\sqrt{\mathrm{NOT}}}^{\left(1\right)}\tilde{\rho }\right).$$ □

As expected, for each level of the syntactical tree of γ, one can determine a corresponding density operator in the qubit-space.

**Lemma** **8.**
*Let $\rho \in D(H_{d}^{\left(n\right)})$. There exists a density operator $\tilde{\rho }\in D\left(H_{2}^{\left(n\right)}\right)$ such that the following conditions are satisfied:*
*(i)* 
*$p\left(Red^{\left(j\right)}\left(\rho \right)\right)=p\left(Red^{\left(j\right)}\left(\tilde{\rho }\right)\right)$;*
*(ii)* 
*$p\left({\;}^{D}{\mathrm{NOT}}^{\left(1\right)}Red^{\left(j\right)}\left(\rho \right)\right)=p\left({\;}^{D}{\mathrm{NOT}}^{\left(1\right)}Red^{\left(j\right)}\left(\tilde{\rho }\right)\right)$;*
*(iii)* 
*$p\left({\;}^{D}{\sqrt{I}}^{\left(1\right)}Red^{\left(j\right)}\left(\rho \right)\right)=p\left({\;}^{D}{\sqrt{I}}^{\left(1\right)}Red^{\left(j\right)}\left(\tilde{\rho }\right)\right)$;*
*(iv)* 
*$p\left({\;}^{D}{\sqrt{\mathrm{NOT}}}^{\left(1\right)}Red^{\left(j\right)}\left(\rho \right)\right)=p\left({\;}^{D}{\sqrt{\mathrm{NOT}}}^{\left(1\right)}Red^{\left(j\right)}\left(\tilde{\rho }\right)\right)$.*



**Proof.** Similar to Lemma 7. □

## 4. Conclusions

In this paper, we consider a semantics which makes essential use of the characteristic holistic features of the quantum-theoretic formalism. Unlike standard compositional semantics, where the meaning of a whole is determined by the meanings of its parts, a holistic model determines the contextual meaning of any subformula.

One can wonder whether the restriction to the space $C^{2}$ is really useful for the aims of quantum computation. A natural “many-valued generalization” of the classical part of quantum computation might assume any $C^{d}$ (where d≥2) as a basic Hilbert space. In this way, elements of the canonical orthonormal basis of $C^{d}$ can be regarded as the truth-values of a many-valued semantics. This gives rise to a natural semantic characterization for a new form of quantum computational logic that has been called “Łukasiewicz quantum computational logic”. As expected, the language of this logic is richer since some basic connectives (like negation and conjunction) turn out to be split into different kinds of connectives. Having in mind the relationships between classical logic and Łukasiewicz-many valued logics, one could expect that **HQCL** is stronger than the fragment **ŁQCL*** of **ŁQCL** whose formulas are expressed in the language of **HQCL**. However, we conjecture this is not the case. This can be explained by recalling that **HQCL** is a very weak form of logic: many important logical arguments are generally violated in the case of **HQCL**. Quantum uncertainties and quantum holism seem to render irrelevant any basic semantic assumption about the number of truth-values.

## Figures and Tables

**Table 1 entropy-23-00735-t001:** Three-valued truth table of the Bertini gate.

Input	Output
|0,0,0〉	|0,0,0〉
$|0,0,\frac{1}{2}\rangle$	$|0,0,\frac{1}{2}\rangle$
|0,0,1〉	|0,0,1〉
$|0,\frac{1}{2},0\rangle$	$|\frac{1}{2},0,0\rangle$
$|0,\frac{1}{2},\frac{1}{2}\rangle$	$|0,\frac{1}{2},\frac{1}{2}\rangle$
$|0,\frac{1}{2},1\rangle$	$|0,\frac{1}{2},1\rangle$
|0,1,0〉	|1,0,0〉
$|0,1,\frac{1}{2}\rangle$	$|1,\frac{1}{2},0\rangle$
|0,1,1〉	|1,1,0〉
$|\frac{1}{2},0,0\rangle$	$|0,\frac{1}{2},0\rangle$
$|\frac{1}{2},0,\frac{1}{2}\rangle$	$|\frac{1}{2},0,\frac{1}{2}\rangle$
$|\frac{1}{2},0,1\rangle$	$|\frac{1}{2},0,1\rangle$
$|\frac{1}{2},\frac{1}{2},0\rangle$	$|\frac{1}{2},\frac{1}{2},0\rangle$
$|\frac{1}{2},\frac{1}{2},\frac{1}{2}\rangle$	$|\frac{1}{2},1,0\rangle$
$|\frac{1}{2},\frac{1}{2},1\rangle$	$|\frac{1}{2},\frac{1}{2},1\rangle$
$|\frac{1}{2},1,0\rangle$	$|\frac{1}{2},\frac{1}{2},\frac{1}{2}\rangle$
$|\frac{1}{2},1,\frac{1}{2}\rangle$	$|\frac{1}{2},1,\frac{1}{2}\rangle$
$|\frac{1}{2},1,1\rangle$	$|\frac{1}{2},1,1\rangle$
|1,0,0〉	|0,1,0〉
$|1,0,\frac{1}{2}\rangle$	$|1,0,\frac{1}{2}\rangle$
|1,0,1〉	|1,0,1〉
$|1,\frac{1}{2},0\rangle$	$|0,1,\frac{1}{2}\rangle$
$|1,\frac{1}{2},\frac{1}{2}\rangle$	$|1,\frac{1}{2},\frac{1}{2}\rangle$
$|1,\frac{1}{2},1\rangle$	$|1,\frac{1}{2},1\rangle$
|1,1,0〉	|0,1,1〉
$|1,1,\frac{1}{2}\rangle$	$|1,1,\frac{1}{2}\rangle$
|1,1,1〉	|1,1,1〉

## Data Availability

Data is contained within the article.

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
