# Peer review of "Quantum Uncertainties and Holism Seem to Render Irrelevant Qudit-Semantics"

_entropy, 2021, doi:10.3390/e23060735_

Round 1

Reviewer 1 Report

This article presents new results on quantum logic, an important and rapidly developing branch of quantum information theory (quantum computing).

This work is mathematical in form, since it is presented as a sequence of  new definitions  corresponding lemmas and theorems. This is close to me, because in my first specialty I am a mathematician. I have carefully studied the proofs of the above lemmas and theorems and I see that they are correct.

The results of the authors are new and I would recommend this work for publication in the journal  “Entropy”  however I have two remarks:

  • in the section References there is clearly not enough reference to some fundamental monograph on quantum logic (now there are a lot of such monographs);

  • This is more of a question than a remark. I tried in vain to understand, but did not understand how the content of the work is directly or indirectly related to the title of the journal “Entropy”. The authors need to explain this somehow. I understand perfectly well that your article was written for a special issue "Quantum Structures and Logics"  and then this question seems superfluous.  Still, I think this question is pertinent.

After eliminating these shortcomings, I support the publication of this article in the journal.  

Author Response

Thanks for suggestions.
We have included some monographs as requested and the relationship with entropy: "In this semantics, the entropy of a formula is the average level of information or uncertainty inherent in the possible truth values."
Entropy is a measurable physical property that is most commonly associated with a state of disorder, randomness, uncertainty.
This notion is used in different fields from classical thermodynamics to the principles of information theory.
Quantum computational logics are based on quantum uncertainty and the truth value associated to a formula is the expected value.

Reviewer 2 Report

The title of the paper is an excellent summary of the purpose of the present paper.

I recommend to the author to quote his previous work {\it Holistic and Compositional Logics Based on the Bertini Gate}, Found. Sci. https://doi.org/10.1007/s10699-020-09703-y
where a similar sentence {\it Quantum uncertainties and quantum holism seem to render irrelevant any basic semantic assumption about the number of truth-values} appears. 
What are the new features of the present contribution compared to his previous work since the topic and several parts of the text look similar.

The paper is good and seems technically correct. The work has potentially a great impact. My only concern is about a possible plagiarism as mentioned above.

Author Response

Thanks for suggestions.
We have included some monographs and the article with the motivation as requested: "In \cite{L20} we showed, as expected, that the compositional logic characterized by the qubit semantics is stronger than the compositional $\L$ukasiewicz quantum computational logic by a counterexample.
The holistic case is a different matter because quantum uncertainties and holism play a fundamental role. "
Lemmas and theorems in this paper serve to support this result which seems counterintuitive: HQCL and the fragment LQCL* characterize the same logic.